

# Rapid drop in the reproduction number during the Ebola outbreak in the Democratic Republic of Congo

Christian L. Althaus

Institute of Social and Preventive Medicine (ISPM), University of Bern, Bern, Switzerland

## ABSTRACT

The Democratic Republic of Congo (DRC) experienced a confined rural outbreak of Ebola virus disease (EVD) with 69 reported cases from July to October 2014. Understanding the transmission dynamics during the outbreak can provide important information for anticipating and controlling future EVD epidemics. I fitted an EVD transmission model to previously published data of this outbreak and estimated the basic reproduction number $R_0 = 5.2$ (95% CI [4.0–6.7]). The model suggests that the net reproduction number $R_t$ fell below unity 28 days (95% CI [25–34] days) after the onset of symptoms in the index case. This study adds to previous epidemiological descriptions of the 2014 EVD outbreak in DRC, and is consistent with the notion that a rapid implementation of control interventions helped reduce further spread.

## INTRODUCTION

The Democratic Republic of Congo (DRC) experienced a confined outbreak of Ebola virus disease (EVD) in rural areas of Équateur province. The first case became ill on 26 July 2014 and the last case started to show symptoms on 4 October 2014, resulting in a total of 69 reported cases (*Maganga et al., 2014*). The index case was a pregnant woman who died on 11 August 2014. A doctor and three health care workers performed a postmortem cesarean section, and all of them became infected and died. In total, 21 cases during the first 24 days of the outbreak had direct contact to the index case (*Maganga et al., 2014*). However, it remains unclear how many of these 21 cases acquired the infection from the index case, and how many infections were generated by subsequent cases.

A better understanding of the transmission dynamics of the 2014 EVD outbreak in DRC can provide useful insights for the anticipation and control of current and future EVD epidemics in rural areas. The average number of secondary infections generated by an infectious index case at the beginning of an outbreak is described by the basic reproduction number $R_0$ (*Heffernan, Smith & Wahl, 2005*). An outbreak can be brought under control once the net reproduction number $R_t$ (also called the effective or instantaneous reproduction number) drops below unity. Several analyses of $R_0$ and $R_t$ for previous EVD outbreaks have given detailed insights into the transmission dynamics and the effectiveness

Corresponding author
Christian L. Althaus,
christian.althaus@alumni.ethz.ch

of control interventions (*Chowell et al., 2004*; *Althaus, 2014*; *Camacho et al., 2014*; *Althaus et al., 2015*).

In this study, I fitted an EVD transmission model to the reported daily numbers of incidence cases during the outbreak in DRC. This allowed me to quantify the transmission rate during this outbreak, provide an estimate of the basic reproduction number $R_0$, and calculate the date at which the net reproduction number $R_t$ fell below one.

## METHODS

I applied the same model fitting procedure that was used to estimate the reproduction number of EVD during the 2014 outbreak in Nigeria (*Althaus et al., 2015*). EVD transmission was described assuming SEIR (susceptible-exposed-infectious-recovered) dynamics using the following set of ordinary differential equations (ODEs):

$$\frac{dS}{dt} = -\beta(t)SI, \tag{1}$$

$$\frac{dE}{dt} = \beta(t)SI - \sigma E, \tag{2}$$

$$\frac{dI}{dt} = \sigma E - \gamma I, \tag{3}$$

$$\frac{dR}{dt} = (1-f)\gamma I, \tag{4}$$

$$\frac{dD}{dt} = f\gamma I. \tag{5}$$

Susceptible individuals, $S$, can get infected by infectious individuals, $I$, at rate $\beta$. They then move through an incubation period ($E$) before becoming infectious individuals, $I$, who either recover or die. $1/\sigma$ and $1/\gamma$ correspond to the average durations of the incubation and infectious period, respectively. The transmission rate was assumed to be constant until time $\tau$, after which it decays exponentially at rate $k$: $\beta(t) = \beta_0 e^{-k(t-\tau)}$.

The daily incidence of onset of symptoms (Data S1) was derived from the study by *Maganga et al. (2014)*. I extended the data set from the time of symptom onset in the last case to the date that the World Health Organization declared the outbreak in DRC to be over (20 November 2014) with zero counts for the number of incident cases. The average durations of incubation and infectiousness were fixed to values from previous outbreaks (*Althaus et al., 2015*). I assumed the outbreak started with the onset of symptoms in the index case in a large susceptible population ($I(0) = 1$ and $S(0) = 10^6 - 1$). Note that the exact number of susceptible individuals does not need to be known for estimating model parameters since the total number of cases is much less than $10^6$.

By numerically integrating the ODEs, the modeled incidence of infectious cases can be calculated as follows:

$$\Delta H(t) = \int_t^{t+1} \sigma E \, dt. \tag{6}$$

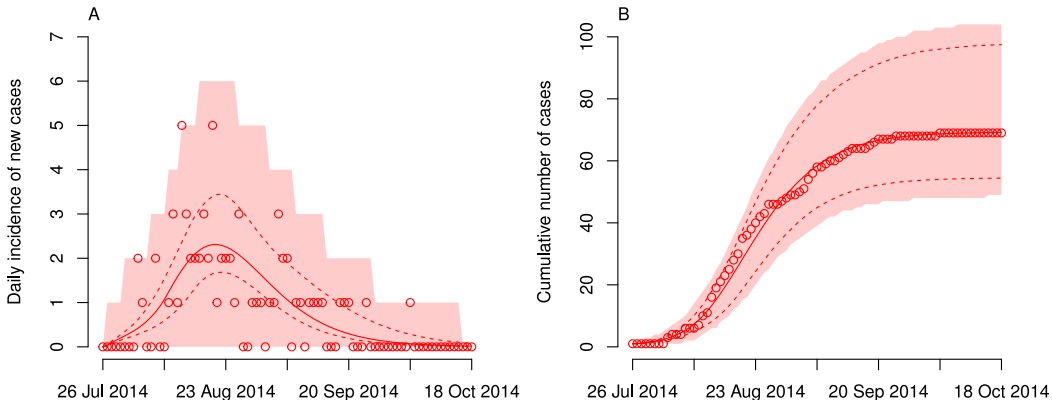

**Figure 1 Dynamics of Ebola virus disease (EVD) outbreak in the Democratic Republic of Congo (DRC).** Model fits of daily incidence (A) and cumulative numbers (B) of cases are shown together with reported data (circles). The best-fit model (solid lines) is given together with the 95% confidence intervals (dashed lines). The shaded areas correspond to the 95% prediction intervals.

Assuming the daily numbers of incident cases to be Poisson distributed (*Camacho et al., 2014*), the likelihood to observe $N(t)$ cases on day $t$ for a given set of model parameters $\theta$ is

$$\mathcal{L}(N(t)|\theta) = \frac{\Delta H(t)^{N(t)} e^{-\Delta H(t)}}{N(t)!}. \tag{7}$$

I minimized the negative log-likelihood to derive maximum likelihood estimates of the following three parameters: the baseline transmission rate $\beta_0$, the time $\tau$ at which transmission starts to drop, and the rate $k$ at which transmission decays. Those parameters were used to calculate the basic reproduction number $R_0 = \beta_0 S(0)/\gamma$, the net reproduction number $R_t = \beta(t)S(t)/\gamma$ and the time at which $R_t$ dropped below unity. I derived simulation based 95% confidence intervals (CIs) for the model curve from $10^4$ bootstrap samples making use of the asymptotic normality of the maximum likelihood estimates (*Mandel, 2013*; *Althaus et al., 2015*). I also constructed 95% prediction intervals (PIs), by simulating a Poisson-distributed daily incidence of cases for each epidemic trajectory. In both cases, I used the 2.5% and 97.5% quantiles from the bootstrap samples at each time point $t$ to construct point-wise intervals.

The results from fitting the deterministic EVD transmission model to the data were compared to estimates of the case reproduction number obtained using the method by *Wallinga & Teunis (2004)* as implemented in the package *R0* (*Boelle & Obadia, 2015*) for the R software environment for statistical computing (*R Development Core Team, 2014*). For this separate analysis, I assumed a gamma-distributed generation time with a mean of 15.3 days and a standard deviation of 9.3 days as reported by the *WHO Ebola Response Team (2014)*. R code files are provided as Supplemental Information and can also be downloaded from GitHub (https://github.com/calthaus/Ebola).

## RESULTS AND DISCUSSION

Fitting the transmission model to the data provided a good description of the EVD outbreak in DRC (Fig. 1). The maximum likelihood estimates of the model parameters

**Table 1 Parameter estimates of the EVD transmission model.** The average duration of incubation and infectiousness were fixed to values from previous outbreaks (*Althaus et al., 2015*).

| Parameter | Description | Value | 95% CI |
|---|---|---|---|
| $R_0$ | Basic reproduction number | 5.15 | 3.95–6.69 |
| $\beta_0$ | Transmission rate (per individual per day) | $0.70 \times 10^{-6}$ | $0.53 \times 10^{-6}$–$0.90 \times 10^{-6}$ |
| $\tau$ | Time at which transmission rate starts to decay (days) | 14.3 | 5.2–23.4 |
| $k$ | Rate at which transmission rate decays (per day) | 0.12 | 0.07–0.23 |
| $1/\sigma$ | Average duration of incubation (days) | 9.31 | – |
| $1/\gamma$ | Average duration of infectiousness (days) | 7.41 | – |

(Table 1) resulted in an $R_0$ of 5.15 (95% CI [3.95–6.69]), which is higher than estimates from other, larger outbreaks (*Chowell et al., 2004*; *Althaus, 2014*; *Camacho et al., 2014*). However, the number is lower than the 21 cases who were reported to be direct contacts of the index case (*Maganga et al., 2014*), suggesting that not all these cases acquired the infection from the same source. It is important to note that the time from onset of symptoms to death in the index patient (16 days) is substantially longer than the average duration of infectiousness in the model (7.4 days). Hence, the number of secondary cases generated by the index case could be more than twice as high as the estimated $R_0$, indicating a potential superspreading event during this outbreak (*Volz & Pond, 2014*; *Althaus, 2015*; *Toth et al., 2015*).

The time $\tau$ at which the transmission rate started to drop was estimated at 14.3 days (95% CI [5.2–23.4] days) after the start of the outbreak (Table 1). This time point is before the death of the index case (16 days after onset of symptoms) and the subsequent cesarean section that lead to four secondary cases. However, the CIs around the estimated time point are wide and are also consistent with a reduction in the transmission rate—possibly due to control interventions—that starts one week after the death of the index case. On 26 August 2014 and 28 days after the EVD outbreak started, the Ministry of Health in DRC notified the WHO of the outbreak (*World Health Organization, 2014*). At that time, the health sector had already mounted a large response to the outbreak, including contact tracing, treatment of patients, infection prevention and control measures. The model indeed suggests that the net reproduction number $R_t$ dropped below unity 27.6 days (95% CI [24.7–33.7] days) after the start of the outbreak (Fig. 2). This time point also roughly coincides with the date at which the modeled daily incidence of new cases is highest (Fig. 1A).

This is the first study inferring the transmission dynamics of the 2014 EVD outbreak in DRC using mathematical modeling. The model is based on an established framework that was applied for the analysis of a limited urban outbreak of EVD in Nigeria (*Althaus et al., 2015*). A major limitation of the model is that it provides a deterministic description of a relatively small outbreak, and assumes homogeneous mixing for a population covering a large rural area. Stochastic models might be better suited to account for the process noise during the early phase of an outbreak and typically result in wider confidence intervals of the estimated parameters (*King et al., 2015*). For example, the outbreak could have

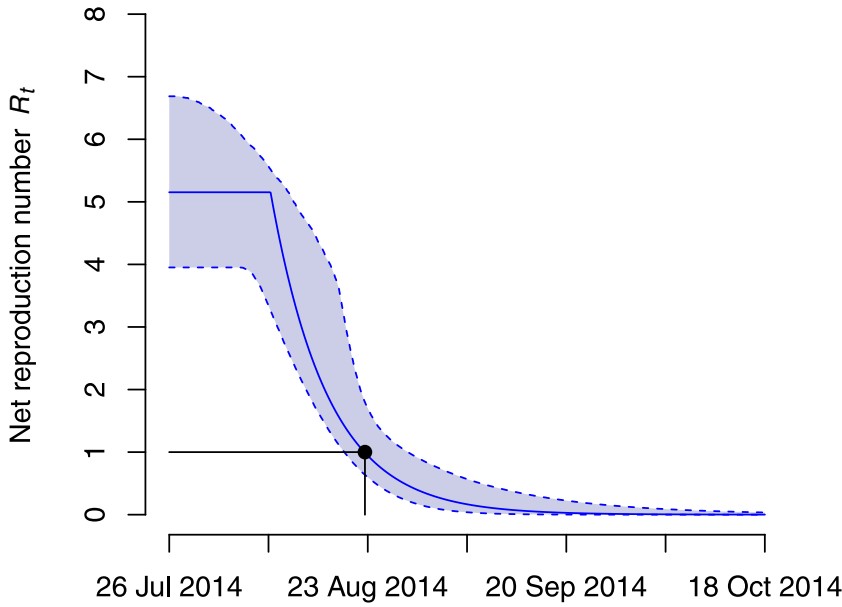

**Figure 2 Net reproduction number $R_t$ during the Ebola virus disease (EVD) outbreak in the Democratic Republic of Congo (DRC).** The maximum likelihood estimates of the net reproduction number $R_t$ (solid line) are shown together with the 95% confidence intervals (dashed lines). The black dot denotes the time at which $R_t$ dropped below unity (27.6 days after the start of the outbreak).

resulted from a single superspreading event by the index case, while the reproduction number might have dropped below unity for all subsequent cases. Therefore, I also analyzed the outbreak using the concept of the case reproduction number by applying a likelihood-based method to calculate the number of secondary infections generated per case by date of onset of symptoms (*Wallinga & Teunis, 2004*). This method does not make any assumptions about how the reproduction number changes with time and can take the individual heterogeneity in the number of secondary cases better into account. The results from this analysis confirmed the findings from the deterministic model (Fig. S1). The estimated case reproduction number of the index case was 5.6 (95% CI [3.0–9.0]). The case reproduction number dropped afterwards but the 95% CIs remained at or above one for the next five cases, highlighting that they might have played an important role in further transmission. The number fell below one for all cases whose symptoms started on 16 August 2014 or later. This suggests that the net reproduction number must have dropped below one within one infectious period (7.4 days) after this date. Indeed, the $R_t$ from the ODE model dropped below one on 23 August 2014 (Fig. 2)

There are a number of further limitations. First, the model does not distinguish between transmission in the community, in health-care settings, and from deceased individuals. Hence, I assumed that the average duration infected individuals remain infectious includes the possibility of transmission from those who died. Second, I assumed that the incubation and infectious period are exponentially distributed, although the original data suggests that those periods can be described by gamma distributions with shape parameters >3 (*Althaus et al., 2015*). However, the arguably more important property determining

the transmission dynamics is the generation time (time between infection in an index case and infection in a secondary case). Fitting a gamma distribution to reported data from the WHO Ebola Response Team (2014) results in a shape parameter of 2.6. With a constant transmission rate during the infectious period, the presented ODE model results in a similar generation time distribution with shape parameter of approximately 2. Third, the model cannot capture the separate contribution of different control interventions, such as contact tracing, case isolation, protection of health care workers, and safe burials in reducing transmission. Fourth, the analysis is restricted to incidence data of symptom onset. Adding an additional variable for incidence of death could result in more accurate parameter estimates.

In summary, this study complements the epidemiological description of the 2014 EVD outbreak in DRC (Maganga et al., 2014) by using mathematical modeling to provide estimates of the reproduction number during the outbreak. It remained unclear to what extend the outbreak was driven by transmission from infected cases other than the index case. The results of this study suggest that the net reproduction number $R_t$ dropped below unity around four weeks after symptom onset in the index case, indicating that a substantial number of subsequent cases might have contributed to further transmission. The time point at which $R_t$ fell below one coincides with the publication of a report stating that an effective response to the outbreak had been set in place by then (World Health Organization, 2014). It was already noted in the study by Maganga et al. (2014) that the fast and effective response, in addition to the remoteness of the area in which transmission took place, was one of the most plausible explanations why the EVD outbreak remained relatively small. While it cannot be ruled out that other factors might have limited further spread of EVD, the findings of this study support the notion that a rapid response resulted in a drop of the reproduction number during the outbreak in DRC.

### Funding

Christian L. Althaus received funding through an Ambizione grant from the Swiss National Science Foundation (project 136737). The funders had no role in study design, data collection and analysis, decision to publish, or preparation of the manuscript.

### Grant Disclosures

The following grant information was disclosed by the author:
Swiss National Science Foundation: 136737.

### Competing Interests

Christian L. Althaus is an Academic Editor for PeerJ.

### Author Contributions

- Christian L. Althaus conceived and designed the experiments, performed the experiments, analyzed the data, contributed reagents/materials/analysis tools, wrote the paper, prepared figures and/or tables.

## Data Availability

Github: https://github.com/calthaus/Ebola.

## Supplemental Information

Supplemental information for this article can be found online at http://dx.doi.org/10.7717/peerj.1418#supplemental-information.

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
