# Peer review of "Rapid drop in the reproduction number during the Ebola outbreak in the Democratic Republic of Congo"

_PeerJ, doi:10.7717/peerj.1418_

## Round 0.1 · original submission · Major Revisions

Both reviewers make a number of salient comments. The most important ones are:
- All methods need to be clearly described
- A deterministic model may be insufficient to infer the conclusions you would like to infer
- There is insufficient evidence to conclude that control measures were the cause of reduced R0

Reviewer 1 ·

Basic reporting

The article is in accordance with most of the basic reporting practices at PeerJ. While the data from the study have been made available, I believe the necessary code for the analysis should also be included with the manuscript for transparency and reproducibility.

Experimental design

The analysis concerns the EVD outbreak in the DRC from the summer of 2014. The study applies methods from published EVD transmission models to this outbreak. But the methods section for this paper is minimal and does not include any attention to idiosyncrasies of this particular outbreak. For instance, the methods used make very strong assumptions that are more appropriate for far larger outbreaks in single large populations. The model used does not adequately reflect the mechanisms and stochastic variability underlying this outbreak, likely leading to biased estimates of parameters (such as R(t)) and their uncertainty. In the paper, “control” refers to any reduction in transmission over time, which contains behavioral change and stochastic noise in addition to any actual interventions. The authors’ estimated timing of “control” may thus simply coincides with the end of the superspreading event that sparked the outbreak, which does not necessarily reflect any real intervention measures. Furthermore, the intent of the paper “to quantify transmission dynamics” is vague and the conclusions--that a rapid response is necessary to stop an outbreak--while likely true, do not follow from the methods/results. These concerns and others are described in greater detail below.

1. While the methods are fully described in the cited references, it would be nice to include a slightly more in-depth description in the current paper. Specifically, the author mentions using an SEIR model for fitting in the current paper and cites the model from Althaus 2015 which, unlike the current paper, contains a death compartment. Also, a simplified description of the prediction interval calculation from Althaus 2015 should be included, because it is not an immediately obvious procedure from the description in the current paper.

2. The SEIR model implemented in the paper seems inadequate to realistically describe the natural history of EVD. In reality, individuals often spend considerable time in the exposed (mode of ~10-11 days) and infectious classes (> 4 days) with ebola virus disease. Yet the current paper uses an ODE model that implies an exponential waiting time spent in each compartment, allowing instant transitions between those classes--ie many individuals in this model become infectious immediately after they’ve been infected. This likely could cause issues with parameter estimation. A boxcar (distributed delay) implementation would give more realistic gamma-distributed latent and infectious periods.

3. I also would refer the author to a paper by King et al. (http://rspb.royalsocietypublishing.org/content/282/1806/20150347), which argues that models of outbreaks should fit stochastic models to incidence data quite convincingly. That paper was discussing how problematic ODE fits are to EVD outbreaks in West Africa, which were far larger. This problem is even further magnified for the far smaller DRC outbreak, which occurred across many small and spatially discrete villages (in contrast to mass action and homogeneity assumptions of an ODE). The lack of consideration of process noise in a deterministic model artificially narrows confidence intervals around estimated parameters, suggesting that the authors’ estimates overestimate their certainty. Furthermore, the model has not been rigorously assessed for fit, which is a useful diagnosis to assess whether it represents an adequate description of the DRC outbreak.

4. The author uses prediction intervals from the model outputs to capture the outbreak data. It is unclear how these intervals are “predictive” since the paper does not claim anywhere to predict an outbreak forward from contemporarily available data. It seems that their purpose is to display the expected amount of observation noise around the true epidemic so that these intervals capture the outbreak data (Fig 1). However, the authors include two forms of error in their prediction intervals: observation error and parameter estimation error. The latter does not make sense to include in this context because this is an observation error only model (no process noise). The author seems to be substituting parameter estimation error for process error--but this is not statistically valid. Well-constructed process error models make prediction intervals that capture the fitted data without incorporating parameter estimation uncertainty. Here, by putting the parameter estimation error into their PIs, they are more likely to capture all the data, appearing to indicate a good model fit. However, it is the MLE epidemic model with observation error around it (the dashed lines) that should contain all the data points if the model is a good fit. In other words, if the MLE model could be the data generating process then it should be able to generate data like the real data. Clearly it cannot; and this suggests problems with the Poisson observation error assumption (too little variance) as well as arises because the model doesn’t have process noise. Thus, while the PIs cover the data generated from the outbreak, they do not fix issues concerning parameter estimation with deterministic models. It seems likely that the confidence intervals surrounding the parameter estimates would be much wider with stochastic implementation and estimation, and therefore give a much wider range of possible outbreak trajectories, fixing issues with the confidence intervals.

Validity of the findings

While the average parameter estimate may not change with stochastic implementation, it is incorrect to attribute all of the model error into the observation process as this gives artificially narrow confidence intervals for the parameter estimate. The prediction intervals capture the data and appear to overcome the model misspecification, but this is misleading since they incorporate extraneous error to do so.

The original goal appears to be: “to quantify the transmission dynamics of this outbreak, provide an estimate of R0, and calculate the date the outbreak was brought under control.“ It is unclear what the author means by quantifying the transmission dynamics of the outbreak other than providing estimates of R(t), a point that should be clarified.

A larger issue is understanding what the author means by calculating the date the outbreak was brought under control. There is no mention in the paper concerning any intervention strategies put into place during the outbreak or the timing thereof. The author implements a generalized control strategy--exponential reduction in transmission rate at an estimated time, tau--but the reduction in transmission could be caused by a number of different scenarios (e.g. behavioral change, interventions, loss of susceptibles in village, etc.). The natural course of an outbreak that has died out necessitates that the R(t) will eventually fall below 1, and the author appears to conflate this observed phenomenon with an inferred intervention to prevent the disease spread.

Furthermore, the authors identify control as having begun shortly after the first index superspreading event. It is quite possible that the reduction in R(t) is neither due to exogenous control nor endogenous behavioral change, but simply stochastic regression to the mean. Ie reported index cases of reported outbreaks are more likely to be superspreaders and thus the individuals they infect are likely to infect fewer cases themselves. The importance of such stochastic effects cannot be evaluated in ODE modeling frameworks.

The author also states: “The study illustrates how mathematical modeling can complement epidemiological descriptions of EVD outbreaks.” It is unclear how the work presented here complements the epidemiological description of Maganga et. al., as they report reproductive numbers for the outbreak stating that exclusion of the index case contacts leads to a reproductive number below the threshold of 1, leading to the outbreak decline.

The author also states: “This highlights the importance of a rapid response to EVD outbreaks in rural areas in order to prevent further spread.” While the reproductive number appears to quickly fall below the threshold of 1, the author has not made any connection to a rapid response in the current outbreak of interest, or how that prevented the rural outbreak from moving to an urban population, as some of the cases did occur in the town of Boende with a population of almost 40,000 people.

Reviewer 2 ·

Basic reporting

The current manuscript does not represent a self-contained unit of publication. Instead of referring to the reader to a prior paper for the details of the methodology, they should be repeated here. This work should include the model, equations used to estimate R_0 and R_t, all MLE parameters, the priors associated with the parameters, and an explanation for how MLE are obtained.

There is a minor concern regarding appropriate reference of prior literature. In the results and discussion the direct reference to Maganga et al 2014 states that they report an R_0 = 21. This is technically correct, but this appears to not be a main conclusion of that paper. First off, they mention this number for R_0 is conditional on all secondary cases resulting from the index case. They do not state whether or not this is expected. Second off, they immediately follow that statement with a report the average number of secondary infections across the whole epidemic and estimate a value of R_t~1 and they show this value is less than 1 when the index case is removed. This second result is the focus of their discussion section as they postulate mechanistic reasons for why the EVD epidemic in DRC has R_t<1 for most time. Overall, the focus of Maganga et al 2014 aligns less with the estimated R_0 = 21 for the epidemic and rather aligns with the idea of the index case being a super-spreading event, something that is mentioned in the current paper.

Experimental design

The methods are not presented adequately and the paper is not reproducible in its current state. A full detail of the methods should be provided. As before, this work should include the model, equations used to estimate R_0 and R_t, all MLE parameters, the priors associated with the parameters, and an explanation for how MLE are obtained.

There is a possible major cause of concern in how the mean infectious and incubation times are determined and subsequently modeled. Althaus et al (2015) reported mean infectious/incubation times based on fits of gamma distributions to two different sets of prior EVD data. Both fits resulted in shape parameters > 3, i.e., unimodal probability densities. Since the model is not reported here it is not clear whether or not the exposed and infectious classes are modeled with multiple stages or “boxcars” to yield gamma distributions or whether single classes are used. The single classes would correspond to non-peaked exponential distributions. If this is the case, the author should justify the modeling decisions in order to resolve any possible contradictions.

Validity of the findings

Some of the conclusions are not appropriately stated. The last sentence of the discussion section in this manuscript should be reworded to be less absolute. This work shows that control as modeled by exponential decay of the spread rate can match the data. But the rapid decrease in disease spread is a property of the data that may or may not be a result of control measures. In particular, the discussion section of Maganga et al 2014 highlight possible reasons (including, but not limited to, control measures) for the decrease of incident cases over time. To suggest that control measures were key for limiting the 2014 EVD outbreak in DRC there should be some reference to control measures implemented by the DRC approximately 2 weeks after the index case.

Additional comments

This manuscript is not fully reviewable until the full methods are presented.

---

## Round 0.2 · Major Revisions

Both reviewers agree that the manuscript is much improved. However, both reviewers had some additional comments. In particular, one of the main points from the previous rounds of review, the question of causality (did early intervention *cause* the rapid decline), remains not fully addressed. Therefore, I would like to ask you to carry out one more round of revisions of your manuscript.

Reviewer 1 ·

Basic reporting

The article is in accordance with the basic reporting practices at PeerJ.

Experimental design

I appreciate the effort that the author has gone through to address my previous concerns. I still have a few comments concerning the experimental design.

In King et al, the author correctly notes that one of the main messages concerned fitting models to cumulative incidence data. However, my initial concerns regarding using the deterministic model are most clearly shown in Figure 1d of the paper (http://rspb.royalsocietypublishing.org/content/282/1806/20150347). Focusing solely on the blue point/error bars, because these indicate the situation of fitting deterministic models to incidence data as is the case in the Author’s paper, one can see that the parameter estimate confidence interval nominal coverage of the true parameter value is still much lower than necessary to be statistically valid (50-75% lower than expected depending on the stochasticity of the outbreak). This suggests that the MLE parameter estimates are artificially narrow, an issue likely to be occurring in the current study as well.

It would be nice for the author to elaborate on the confidence intervals for Figure 1. Are these lines representative of actual epidemic trajectories, or are they simply the 0.025 and 0.975 quantile of values for each time point? If they are epidemic trajectories, how were the quantiles determined chosen?

Is there a reason the value and 95% CI for transmission rate are missing in the table?

Validity of the findings

In terms of the validity of the findings, it is likely that the average parameter estimate may not change with my main concerns above (Though based on the King paper, I would change the author’s words to “are likely” artificially narrow, rather than “potentially” in their statement).

The main conclusions/purpose of this paper are still unclear. Experimental design issues aside, the author has simply fit a standard model to data from a specific outbreak. While it may be true that R0 fell below unity 4 weeks after the start of the outbreak, it is unclear what this adds to our knowledge of ebola virus disease (EVD) transmission. Any outbreak that has ended will fall below unity eventually, and the fact that this time period corresponds with the WHO report could be a coincidence or something real. However, from the look of Figure 2, it appears the reproduction number was falling steeply, significantly before that time point and therefore before significant interventions were put into place. This would suggest an alternative hypothesis that the index case infected a disproportionate number of people, and the subsequent infections resulted in few transmission events unrelated to interventions. This same dynamic is also apparent in the secondary analysis the author completed using the Wallinga method. The reproduction number decreases significantly early on, and could potentially (by an eyeball of the 95% CI from Supplementary figure 1) have fallen below unity in early august, which is well before the WHO reported interventions, and would correspond to roughly one generation of transmissions.

It is also unclear how this complements the maganga et al initial epidemiological description of the outbreak as stated by the author. One might say that it provides a nice quantitative visualization of the reproduction number over time, however, the results from this study are not related back to the maganga study at all. In fact, the maganga study stated that without the initial superspreading event, the overall R0 was well below unity, which appears to already answer the question of transmission dynamics over time, and is consistent with my comments in the above paragraph.

Finally, the author states: “The study by Maganga et al. (2014) already noted that the fast and effective response was likely one of the major reasons that the EVD outbreak remained relatively small. The present study further highlights the importance of a rapid response to EVD outbreaks in rural areas in order to prevent further spread.“ I believe this statement is extremely misleading for a number of reasons:

The Maganga paper highlights 5 reasons that could have lead to the outbreak being small, of which a fast response was only one.

The current study does not look at anything related to rural outbreaks compared to others, and in fact uses methods more suitable for larger more homogenous urban populations.

There is no analysis related to the extent in which this outbreak would have grown in a non rural environment. Would it have been bigger? There were cases in a village of 12,000 people, does that count as a city?

The current study has not highlighted the importance of a rapid response, but rather that the reproductive number fell rapidly early on in the outbreak. I don’t think anyone would argue against the importance of a fast response to an outbreak, but the current study doesn’t investigate the influence that had on this outbreak.

Reviewer 2 ·

Basic reporting

Minor revisions are required to fix typos that can confuse the reader. The major change is the notation for the incidence of infectious cases $\Delta I(t)$ should be changed. Current notation suggests we are tracking the overall change in the number of infectious between time points whereas only the increase due to individuals transitioning from exposed to infected is tracked.

I attached a manuscript with other suggested edits to help improve clarity.

Experimental design

This paper leverages previously published methods, epidemiological data, and computer packages for previously unexplored case data from a modeling perspective. Most assumptions are explicitly mentioned and justified.

One caveat is the author chooses to use an SEIR model, which does not explicitly model disease spread from deceased individuals. Since the index case presumably causes disease spread during the post-mortem cesarian section, the infectious class can be interpreted as including both living and deceased infectious individuals. Currently, the text states "infectious individuals I that recover or die at rate." This should be reconciled. Similarly, how transmission from deceased individuals is interpreted within the current model or justifiably not included should be addressed either in the methods or discussion.

Validity of the findings

The results are based on previously published material and the provided r scripts reproduce the figures and results. Robustness of the results is provided by comparison to alternative methods (Wallinga and Teunis 2004).

It may be valuable to identify robustness of the results with respect the data itself. The incident data was appended with 0 incidences up until the DRC was declared free of ebola. The amount of time from the last case to the declaration is over 1/4 of the entire time frame. A robustness analysis by comparing results between data with different numbers of appended 0s may be helpful in determining the robustness of results. It's my personal opinion that this is not completely necessary as the expectation is R_t will fall below 1 near the peak of index cases regardless of the number of appended 0s, however the editor may deem this important to address.

Additional comments

Overall, this is a much improved paper in terms of clarity, robustness, and justification of claims.

Annotated reviews are not available for download in order to protect the identity of reviewers who chose to remain anonymous.

---

## Round 0.3 · accepted · Accept

Thanks for carrying out a careful second round of revisions.